# Oral Administration of Ethanolic Extract of Shrimp Shells-Loaded Liposome Protects against Aβ-Induced Memory Impairment in Rats

**DOI:** 10.3390/foods11172673

**Published:** 2022-09-02

**Authors:** Zulkiflee Kuedo, Lalita Chotphruethipong, Navaneethan Raju, Ratchaneekorn Reudhabibadh, Soottawat Benjakul, Pennapa Chonpathompikunlert, Wanwimol Klaypradit, Pilaiwanwadee Hutamekalin

**Affiliations:** 1Division of Health and Applied Sciences, Faculty of Science, Prince of Songkla University, Hat Yai, Songkhla 90110, Thailand; 2Department of Food Science, Faculty of Science, Burapha University, Mueang Chonburi, Chonburi 20131, Thailand; 3International Center of Excellence in Seafood Science and Innovation, Faculty of Agro-Industry, Prince of Songkla University, Hat Yai, Songkhla 90110, Thailand; 4Faculty of Science and Technology, Hatyai University, Hat Yai, Songkhla 90110, Thailand; 5Expert Center of Innovative Health Food and Biodiversity Research Centre, Thailand Institute of Scientific and Technological Research, Khlong Luang, Pathum Thani 12120, Thailand; 6Department of Fishery Products, Faculty of Fisheries, Kasetsart University, Chatuchak, Bangkok 10900, Thailand

**Keywords:** ethanolic extract, shrimp shells, liposome, amyloid-beta, memory impairment

## Abstract

Alzheimer’s disease is characterized by a progressive loss of memory and cognition. Accumulation of amyloid-beta (Aβ) in the brain is a well-known pathological hallmark of the disease. In this study, the ethanolic extract of white shrimp (*Litopenaous vannamei*) shells and the ethanolic extract-loaded liposome were tested for the neuroprotective effects on Aβ_1–42_-induced memory impairment in rats. The commercial astaxanthin was used as a control. Treatment with the ethanolic extract of shrimp shells (EESS) at the dose of 100 mg/kg BW showed no protective effect in Aβ-treated rats. However, treatment with an EESS-loaded liposome at the dose of 100 mg/kg BW significantly improved memory ability in Morris water maze and object recognition tests. The beneficial effect of the EESS-loaded liposome was ensured by the increase of the memory-related proteins including BDNF/TrkB and pre- and post-synaptic protein markers GAP-43 and PSD-95 as well as pErk1/2/Erk1/2 in the cortex and hippocampus. These findings indicated the neuroprotective effects of the EESS-loaded liposome on Aβ-induced memory impairment in rats. It produced beneficial effects on learning behavior probably through the function of BDNF/TrkB/pErk1/2/Erk1/2 signaling pathway and subsequently the upregulation of synaptic proteins. The present study provided evidence that the neuroprotective property of the ESSE-loaded liposome could be a promising strategy for AD protection.

## 1. Introduction

Alzheimer’s disease (AD) is mostly linked with the accumulative formation of amyloid-beta (Aβ) peptides in the brain with progressive loss of memory and cognitive function [1]. Studies have suggested that oligomeric Aβ plays an important role in mediating neuronal damage and synaptic dysfunction in AD [2]. A study showed that an animal model of AD exhibited impairment in memory and cognitive ability in the Morris water maze test (MWT) and object recognition test (ORT). Moreover, the impairment was correlated with the alteration of synaptic proteins in the brain [3]. In addition, a previous study reported that the activation of tropomyosin-related kinase B (TrkB) by brain-derived neurotrophic factor (BDNF) played a pivotal role in plasticity and memory formation. This activation required multiple intracellular signaling pathways including the phosphatidylinositol 3-kinase (PI3K)/protein kinase B (Akt) and the extracellular-signal-regulated kinase (Erk) [4].

Learning and memory are linked with the neuronal brain functions and plasticity. In addition, the formation of synaptic plasticity is associated with the activation of the BDNF/pErk1/2/Erk1/2 signaling pathway [5]. On the other hand, downregulation of BDNF expression caused cognitive and memory impairment in BDNF-knockout mice. Moreover, blocking the Erk1/2 downstream signaling pathway inhibited neurite formation [6]. Furthermore, activation of the BDNF/Erk1/2 signaling pathway promoted the expression of synaptic proteins such as GAP-43 and PSD-95. Upregulation of GAP-43 and PSD-95 stimulated neurite outgrowth, synaptic formation, and plasticity which can develop learning and memory ability [7,8]. Prior studies showed that the accumulation of Aβ in AD brain induced impairment of this signaling pathway [9,10]. Moreover, the infusion of Aβ_1–42_ induced impairment of growth-associated protein-43 (GAP-43) and postsynaptic density protein-95 (PSD-95) in an animal model of AD [11,12].

Shrimp processing in the seafood industry generates a large amount of shrimp waste, including head and shells [13]. However, a study showed that shrimp waste contains several carotenoids such as zeaxanthin, β-carotene, including astaxanthin (AST) which have various health benefits and neuroprotective properties [12,14,15]. AST is a xanthophyl carotenoid found in several marine animals, and is the major abundant and commercial source of the microalgae *Haematococcus pluvialis* [16]. AST exhibited neuroprotective property by promoting neuronal functions, neural plasticity and cognitive function [15,16]. However, a high amount of polyunsaturated fatty acids and the carotenoids in shrimp extract are prone to oxidative reaction when exposed to oxygen, light, and heat [17]. Microencapsulation is a widely used methodology to protect active agents from adverse conditions and improve stability. The lipid bilayer of liposome particles is widely used for the drug-delivery system to improve stability and bioavailability of hydrophilic and hydrophobic drugs. Previously, it was reported that liposomes can entrap lipophilic drugs and improve their solubility in aqueous body fluids [18,19]. Herein, this study aimed to investigate the neuroprotective effect of ethanolic extract from shrimp shells (EESS) by using liposome encapsulation. The study was conducted on Aβ-induced memory impairment in rats.

## 2. Materials and Methods

### 2.1. Extraction of EESS and RP-HPLC Analysis of AST Content in the EESS

Shrimp shells were scraped free of meat, rinsed with running tap water, and air dried at room temperature. The shells were blended with food grade ethanol in the ratio of 1:2 (*w*/*v*). The mixture was filtered and further evaporated to remove ethanol by using Büchi rotary evaporator (Büchi Labortechnik AG, Flawil, Switzerland) at 40 °C. The RP-HPLC was performed to measure AST content in the extract. The analyses were carried out in isocratic conditions by using BDS hypersil C18 column (150 mm × 4.6 mm) packed with 5-mm diameter particles. The mobile phase was methanol/acetonitrile (50:50), the flow rate was 1 mL/min, and the detection wavelength was set at 480 nm. The chromatographic peaks of AST in the extract were identified by comparing the peaks of AST standard and expressed AST content as mg/g extract.

### 2.2. Preparation of EESS-Loaded Liposome

The EESS-loaded liposome was prepared by a thin-film hydration method described by Pintea et al. [20] with some modifications. Briefly, 5 g of soybean lecithin containing about 70% of phosphatidylcholine (GmbH, Ludwigshafen, Germany) was dissolved in ethanol and heated at 45 °C to complete dissolution. Subsequently, 5 g of the EESS was added to phospholipid solution and stirred on a hot plate at 40 °C for 5 min. The ethanol was then removed under reduced pressure at 40 °C by using rotary evaporation to get the lipid film. The film was then hydrated with 100 mL of distilled water to obtain liposomal suspension. Finally, the liposomal particles were downsized by using a water bath sonicator (Crest 1875D, Crest Ultrasonics Corp., Trenton, NJ, USA) and further stored in brown glass vials at 4 °C until used.

### 2.3. Characterization of EESS-Loaded Liposome

#### 2.3.1. Encapsulation Efficiency (EE)

The suspension of EESS-loaded liposome was lyophilized by using a freeze dryer (CoolSafe 55, ScanLaf A/S, Lynge, Denmark). The obtained power was determined for EE as detailed by Gulzar & Benjakul [21]. Surface oil (SO) was recovered by mixing 2 g powder with 15 mL of hexane, followed by filtration through a Whatman No. 1 filter paper. Thereafter, hexane (20 mL) was used for mixing the collected powder three times. The filtrate was placed in a round bottom flask and the evaporation of solvent was performed until the weight was constant. For total oil (TO) recovery, 2 g powder of liposome was dissolved in the mixture solution containing 5 mL of 0.88% (*w*/*v*) KCl solution, 25 mL of methanol and 50 mL chloroform. The mixtures were subsequently homogenized for 5 min at 15,000× *g* and transferred to a separating funnel. The chloroform phase was pooled and removed by evaporation. The EE of liposome was calculated as follows:EE = [(TO − SO)/TO] × 100.(1)

#### 2.3.2. Particle Size, Polydispersity Index and Zeta Potential

The EESS-loaded liposome and blank liposome were freshly prepared in liquid form. Particle size, polydispersity index, and zeta potential of liposomes were measured by using a dynamic light-scattering technique by using a ZetaPlus zeta potential analyzer (Brookhaven Instruments Corporation, Holtsville, NY, USA) [22].

#### 2.3.3. Morphology Study

The EESS-loaded liposome and blank liposome were visualized for morphology by using transmission electron microscopy (TEM) as described by Li et al. [23].

### 2.4. Animals

Male Wistar rats (200–250 g) were housed in a room with temperature maintained at 23 °C and a 12/12-h light/dark cycle. The animals had free access to food and water. The protocol of this study was approved by the Institutional Animal Care and Use Committee, Prince of Songkla University (Ethic no. MOE 0521.11/681).

### 2.5. Experimental Design

Rats were randomly divided into seven groups which shown in following Table 1:

All administrations were fed daily by using an oral gavage for 4 weeks (Table 1). Subsequently, animals received Aβ infusion except the sham group, which received aCSF instead of Aβ. Aβ_1–42_ peptide was reconstituted in aCSF to a final concentration of 1 mg/mL followed by incubation at 37 °C for 4 days to induce aggregation. Rats were anesthetized with 40 mg/kg thiopental sodium by i.p. injection. The aggregated Aβ (10 μL) was infused by intracerebroventricular (i.c.v.) injection into bilateral ventricles at coordinates 1.2 mm posterior to bregma, 1.5 mm lateral to sagittal suture, and 4.0 mm ventral to the skull, according to the stereotaxic atlas [24]. By 3 days after surgery, the feedings were continued for 4 weeks and the behavioral tests including MWT and ORT were performed at 2-week and 4-week periods, respectively, after Aβ induction. After behavioral experiments, rats were anesthetized with 120 mg/kg thiopental sodium by i.p. injection and the cortex and hippocampal tissues were collected for further protein expression study (Figure 1).

### 2.6. Morris Water MAZE Test

MWT was used to evaluate spatial memory performed according to the method described by Morris [25] with some modifications. The water maze consisted of a circular pool (diameter 150 cm and high 60 cm) that was divided into four quadrants and filled with water that the temperature was maintained at 23 °C. One of the quadrants was fixed as the target quadrant with an escape platform (10 cm in diameter). The visual cues in different color and shape were attached at the wall of the pool. During the training trial, the pool was filled with water, and the platform was about 2 cm above the water surface. Each rat was habituated and trained for 4 swimming trials/day with 5 min of interval for 4 consecutive days. The rat was placed in the water, facing the wall of the pool at the starting point, and allowed to find the platform within 60 s. The animals which failed to locate the platform in 60 s would be guided to the platform. After catching the platform, the animal was allowed to stay on it for 30 s to observe and memorize the visual cues and platform location. During the test trial, the water was made opaque with non-toxic powder and filled up the pool until submerging the platform with 2 cm below the water surface. The rat was placed in the pool and the time required to reach the hidden platform was recorded as escape latency. Twenty-four hours h after the test trial, the platform was removed from the target quadrant and the probe trial was conducted. Each rat was allowed to swim for 60 s, and the time spent in the target quadrant was also recorded (Figure 2).

### 2.7. Object Recognition Test

The ORT was performed to assess the cognitive and memory behaviors as described previously [26]. This test was conducted in the open field, to which the animal was first habituated for 10 min. In the training session, each rat was allowed to explore to two identical objects placed in the middle of open field for a duration of 5 min. The rat was then put back in the cage. After the training session, one of the familiar objects was removed and replaced by the novel object and the exploration procedure was repeated at 1 h and 24 h (Figure 3). The rat’s preference for a novel object was calculated and expressed as the percentage of preference index (PI) compared to a familiar object according to the following formula. To remove the presence of odor cues, the apparatus and the objects were cleaned with 70% ethanol before each trial. We have
PI = [T2/(T1 + T2)] × 100,(2)
where T1 and T2 are exploration time to old object and novel object respectively.

### 2.8. Western Blot Analysis

Brain tissues were lysed with RIPA buffer containing a cocktail of protease inhibitors and phosphatase inhibitors. Lysate was centrifuged at 14,000 rpm at 4 °C for 15 min and the supernatant was collected. The total protein concentration was determined by using the Bradford assay (Bio-rad Laboratories, Hercules, CA, USA). Equal amounts of protein were loaded on 10% SDS-PAGE gel and then transferred onto a PVDF membrane (Millipore, Temecula, CA, USA). After blocking with 5% skim milk or 3% BSA, the membrane was incubated overnight at 4 °C with primary antibodies against BDNF, TrkB, PSD-95, Erk1/2, pErk1/2 (Abcam, Cambridge, UK), and GAP-43 (Santa Cruz Biotechnology, Santa Cruz, CA, USA). After washing with TBS-T, the membrane was incubated with the proper secondary antibody conjugated to HRP for 2 h at room temperature. Interested protein signals on the membrane were detected by using SuperSignal West Pico chemiluminescence substrate (Thermo Fisher Scientific, Rockford, IL, USA). The signal intensity of each band on the film was quantified by using ImageJ software (Java 8) and normalized to the corresponding β-actin (Cell Signaling Technology, Danvers, MA, USA) as an internal control.

### 2.9. Data Analysis

The results are presented as mean ± S.E.M. Statistical differences among experimental groups were analyzed by one-way analysis of variance (ANOVA), followed by Tukey’s post-hoc test. Differences with *p* < 0.05 were considered to be statistically significant.

## 3. Results

### 3.1. AST Content in Ethanolic Extract from Shrimp Shells

Previously, shrimp shell extract containing AST content was found to exhibit neuroprotective property [27]. Therefore, the present study was designed to examine the effects of the bioactive compound on neuroplasticity in AD animals. Ethanolic extraction of white shrimp (*Litopenaeus vannamei*) shells gave the extraction yield of 11 g/kg shells. In addition, AST content in the extract was determined by a reverse-phase high-performance liquid chromatography (RP-HPLC) analysis. The chromatogram of the AST standard was shown in Figure 4a with the peak at retention time of 2.072 min. The chromatogram of EESS presented the one major peak at retention time of 2.067 min (Figure 4b) which corresponded with AST standard. According to the peak area of AST standard and EESS, the content of AST in the extract was 2.53 ± 0.11 mg/g extract.

### 3.2. Encapsulation Efficiency (EE), Particle Size, Zeta-Potential and Transmission Electron Microscopy (TEM) Study of EESS-Loaded Liposomes

As presented in Table 2, the EE of a liposome loaded with EESS was 55.34%. In addition, the particle size of blank liposome and EESS-loaded liposome was 587.20 ± 12.90 and 1009.40 ± 38.8 nm, respectively. Moreover, polydispersity index (PDI) of blank liposome and EESS-loaded liposome was 0.40. The morphology of blank liposome and EESS-loaded liposome is illustrated in Figure 5a,b, respectively. Overall, both samples had a spherical shape. EESS-loaded liposome showed a larger size than the blank liposome, which was consistent with the result of particle size (Table 2).

### 3.3. EESS-Loaded Liposome Improved Memory and Cognitive Ability in the Memory-Related Behavioral Tests

To examine neuroprotective effects of EESS-loaded liposome in Aβ-induced cognitive and memory impairment, rats were pre-treated with EESS and EESS-loaded liposome at the dose of 100 mg/kg BW for 8 weeks. The efficacy of both forms of EESS was compared with control AST-commercial. Aβ-treated rats showed a significantly prolonged latency to reach the escape platform compared to that of the sham group. Treatment with the non-encapsulated EESS showed no different effect compared to that of the Aβ group. However, the prolonged escape latency induced by Aβ was significantly reduced by administration of EESS-loaded liposome compared to that of Aβ group. The effect of EESS-loaded liposome was similar to that of the AST-commercial (Figure 6a). Furthermore, time spent in the target quadrant during the probe trial was significantly decreased in Aβ-treated rats compared to the sham group. On the other hand, rats administered with EESS-loaded liposome showed a significant increase of time spent in the target quadrant compared to Aβ group (Figure 6b). The present results expressed that Aβ infusion successfully induced spatial memory deficits and treatments with EESS-loaded liposome potentially attenuated memory deficits in AD rats.

Furthermore, the effect of EESS-loaded liposome on recognition was determined by the ORT. In the training session, there was no difference in the preference index (PI) between two similar objects among the experimental groups. However, during the retention session, Aβ-treated rats displayed cognitive impairment by a significant decrease in the percentage of PI to novel object compared to the sham group. In contrast, rats pretreated with EESS-loaded liposome showed a significant increase in PI to the novel object compared with the Aβ group. Moreover, the neuroprotective effect of the EESS-loaded liposome was similar to that of AST-commercial and higher than that of EESS (Figure 7a,b). These findings indicated that encapsulation with liposome could improve the beneficial effects of ethanolic extract from white shrimp shells in Aβ-induced memory and cognitive impairment in rats.

### 3.4. EESS-Loaded Liposome Increased the Levels of BDNF and TrkB in AD Rats

Previously, BDNF and its receptor, TrkB, appeared to play a pivotal role in regulating synaptic plasticity and memory process [28]. To determine whether BDNF and TrkB associated with Aβ-induced cognitive and memory impairment, the cortex and hippocampus of rats were examined by using Western blot analysis. Rats treated with Aβ_1–42_ alone were found to have significantly decreased BDNF and TrkB expression in the cortex and hippocampus compared to sham control group. The pre-treatment with EESS-loaded liposome and AST commercial exhibited the reverse Aβ effects on BDNF and TrkB reduction without any effect on non-encapsulated EESS and blank liposome-treated group (Figure 8). These results indicated that EESS-loaded liposome could protect Aβ-induced synaptic plasticity and memory loss through BDNF and TrkB.

### 3.5. EESS-Loaded Liposome Ameliorated Aβ-Suppressed GAP-43 and PSD-95 Synaptic Plasticity Proteins

Because GAP-43 and PSD-95 are considered as important markers of synaptic plasticity [8,29], EESS-loaded liposome attenuated Aβ-induced cognitive and memory deficits via increased synaptic plasticity proteins was performed. Thus, expressions of GAP-43 and PSD-95 were assessed (Figure 9). Consistent with the observation above, the data presented here demonstrated that the expression levels of GAP-43 and PSD-95 were significantly decreased in Aβ and blank liposome-treated groups. On the other hand, EESS-loaded liposome and AST-commercial significantly induced the expression of GAP-43 and PSD-95. These findings indicated that EESS-loaded liposome attenuated Aβ-induced synaptic plasticity impairment.

### 3.6. EESS-Loaded Liposome Suppressed Aβ-Induced Cognition and Synaptic Plasticity Deficits via Erk Signaling Pathway

To further support the hypothesis that EESS-loaded liposome-mediated protection is via the activation of the Erk signaling pathway, the expression of Erk and phosphorylation of Erk (p-Erk) were examined. As shown in Figure 10, Aβ and blank liposome-treated groups significantly decreased p-Erk/Erk activity in the cortex and hippocampus compared to the sham control group, whereas the significant increase of p-Erk/Erk expression was found in rats treated with EESS-loaded liposome and AST-commercial, in comparison to the Aβ-treated group. The data suggested EESS-loaded liposome was able to rescue the Aβ-induced cognition and synaptic plasticity deficits.

## 4. Discussion

AD is a neurodegenerative disease and the most common type of memory and cognitive problem among older adults. One of the pathological hallmarks of AD is accumulation of Aβ in the brain [1]. Today, there is no specific treatment for AD. Searching for natural products to prevent or inhibit the progression of AD has gained particular attention recently [30]. Previous studies reported that shrimp waste including head and shells contained several bioactive compounds and carotenoids such as b-carotene and AST [31]. b-carotene and AST are orange pigment with various health benefits such as anti-cancer and anti-neurodegenerative disease effects, but they is unstable and sensitive to heat, light, and oxygen and have poor water solubility [14,15,17]. Due to the limitations of these natural compounds, encapsulation has been the technique to improve stability and bioavailability [14]. In addition, prior evidence noted that liposome protects the degradation of active substances from oxidation by heat and light [17]. Moreover, the liposome improves the cell penetration of the substances and is therefore usually used to deliver drugs in medicine [32]. Thus, the present study attempted to investigate the beneficial effect of the extract from white shrimp shells loaded with liposome in animal model of AD. In general, EE is a crucial parameter indicating the amount of EESS encapsulated in the liposome. The EESS-loaded liposome showed lower EE than that of liposome loaded with shrimp oil prepared by using microfluidization process (75.18%) [21]. However, different processes used for preparation might affect the different EE. After loading liposome with EESS, the particle size was differently increased as compared to blank liposome. This was probably caused by the replacement of lipid bilayer space of liposome with EESS containing polar lipids such as AST found in EESS [33]. Some hydrophobic domains of AST might be embedded between lipid bilayer of liposome via hydrophobic–hydrophobic interaction, whereas some hydrophilic regions of AST, especially the OH group, could insert in core of liposome via H-bonding. Apart from EE and particle size, both samples had negative surface charge (less than −30 mV), which reflected stabilization of liposomes [22]. Nevertheless, the decreased negative surface charge of liposomes was found after loading with EESS. Some un-encapsulated EESSs, particularly those rich in hydrophobic groups, were more likely localized surrounding the liposome. Thus, the repulsion of surface charge between phosphate head groups of liposomes and those with a negative charge took place. When considering PDI, both empty and EESS-loaded liposomes had high uniformity of dispersion as indicated by smaller PDI values than 0.7 [34]. Thus, the liposome prepared by using the thin-film hydration method could be effective for the encapsulation of EESS, which resulted in the liposome having high EE, stability, and homogeneity. The TEM investigation revealed a spherical shape with double-layer morphology of the EESS-loaded liposome and blank liposome. The liposome loaded with EESS had a larger size than the empty one, indicating that EESS was loaded in the vesicle. Similar to the result of particle size, the increased particle size was observed for the liposome loaded with EESS. Based on the RP-HPLC result, EESS containing AST could be encapsulated in the lipid bilayer of the liposome via hydrophobic–hydrophobic interaction [33]. Moreover, some hydrophilic parts of AST, especially OH group, might bind to polar lipid bilayer of wall via H-bonding [35]. This presumably led to the enhanced size of liposome after loading with EESS.

Next, EESS and EESS-loaded liposome were used to investigate the neuroprotective effect in AD rats. The behavioral assessment revealed that Aβ-treated rats exhibited impairment of spatial memory in MWT at 2 and 4 weeks after Aβ infusion. Moreover, Aβ-treated rats also showed a decrease in spontaneous exploring behavior to novel object in ORT at both experimental time points. This behavioral alteration of Aβ-treated rats indicated that infusion of Aβ effectively induced memory and cognitive impairment in rats. A previous study reported that accumulation of Aβ in the brain is an important pathological hallmark of AD by which Aβ plaque can induce neuronal damage, synaptic loss, and cognitive decline [36]. The present study found that administration of non-encapsulated EESS did not exhibit protective effects against Aβ-induced memory deficits. However, administration of EESS-loaded liposome in Aβ-treated rats significantly improved spatial memory parameters in MWT and PI to novel object in ORT. In addition, the effect of the EESS-loaded liposome on memory ability of these behavioral tests was similar with a positive control, AST-commercial. MWT is a well-known test of spatial learning and long-term memory and has been shown to be highly associated with spatial memory and hippocampal function [37]. Moreover, ORT is a non-spatial memory and related with the function of the perirhinal cortex [38]. Based on the recent findings and prior evidence, the present study suggested that the neuroprotective effects of EESS-loaded liposome on memory ability might be involved in neuronal brain function of cortex and hippocampus. A previous study demonstrated that BDNF via activation of TrkB receptor plays a critical role in synaptic plasticity and memory formation. BDNF can activate TrkB at pre- and postsynaptic sites in promoting synaptic function to increase the docked vesicles at the active zone of synapses and promotes the presynaptic release. BDNF via Erk signaling can upregulate surface expression of APMA receptors to increase excitatory transmission at postsynaptic cites [39]. Moreover, the BDNF/TrkB/Erk signaling pathway is also involved in neuronal survival and the formation of dendritic spines [5,40]. Studies reported that BDNF/TrkB activation is involved in the expression of GAP-43 and PSD-95 [41,42]. The presynaptic protein GAP-43 is physiologically involved in axonal growth and regeneration as well as synaptogenesis, which is wildly expressed in the prefrontal cortex and hippocampus [7]. Phosphorylation of GAP-43 promoted neurite outgrowth and enhanced vesicle cycling resulting in an increase of neuronal plasticity [29]. In addition, PSD-95 is a scaffolding postsynaptic protein enrolling in synaptic formation and plasticity [8]. Recently, the neuroprotective activity of EESS-loaded liposome was ensured by measuring the expression of those memory-related proteins in cortex and hippocampal tissues. The i.c.v. infusion of Aβ_1–42_ effectively reduced the expression of BDNF and TrkB. Moreover, a presynaptic protein, GAP-43, and a postsynaptic protein, PSD-95, as well as p-Erk1/2/Erk1/2 expression is also reduced in the cortex and hippocampus. On the other hand, EESS encapsulated in the liposome significantly increased the expression of BDNF, TrkB, p-Erk1/2/Erk1/2 and GAP-43 and PSD-95 in the cortex and hippocampus This is similar to the results achieved with AST-commercial. The increase of these memory-related proteins was correlated with the improvement of memory in the behavioral studies. A previous study reported that improvement of memory ability in rats with post-stroke rehabilitation was associated with the increase of BDNF/TrkB, as well as GAP-43 and PSD-95 [43]. Moreover, a study reported that feeding shrimp AST could protect memory impairment and brain tissue damage in Aβ-treated rats [27]. In addition, feeding AST emulsifier at a dose of 0.02% in aging rats could protect hippocampal damage and upregulation of BDNF expression [44]. Thus, the present study suggested that EESS-loaded liposome possibly exhibited neuroprotective effects against Aβ-induced memory deficits by promoting the BDNF/TrkB/Erk signaling pathway. This activity might further stimulate the expression of the presynaptic protein GAP-43 and the postsynaptic protein PSD-95 in the cortex and hippocampus. However, treatment with non-encapsulated EESS showed no potential effect on the expression of these proteins similar to the behavioral studies. A prior study reported that shrimp oil contains a large amount of polyunsaturated fatty acids, b-carotene, and AST which are prone to oxidative reaction [45]. The low beneficial effects of EESS may be due to poor water solubility and low stability of EESS’s bioactive compounds on exposure to oxygen, heat, and oxygen as well as the harsh conditions of the gastrointestinal tract [17]. However, a prior study reported that low stability and bioavailability of active agents could be improved by liposomal encapsulation [46]. The liposome can protect active compounds from harsh conditions and slow down the rate of enzymatic degradation in the gastrointestinal tract [47]. The lipid composition of the liposome can stimulate the production of chylomicrons in enterocytes and promote drug transportation into the lymphatic system [32]. Previously, shrimp AST loaded with liposome that contains 70% of phosphatidylcholine was found to exhibit higher than 90% of cellular uptake across Caco-2 cells [35]. The development of AD is a multifactorial process, including aggregation of amyloid plaques. The misfolded proteins that aggregate all induce oxidative stress [48]. A recent study also demonstrated that astaxanthin plays an important role in antioxidant which mitigates an inflammation, progression, and exacerbation of AD [49]. These indicated that the antioxidant effect of EESS-liposome may also play an important role in neuroprotection. Thus, the antioxidant effect of EESS-liposome may be one of mechanisms contribute to protect the development of AD. According to our study, we found that EESS without liposome exhibited no significance of neuroprotective effects on memory ability. In addition, treatment of blank liposome without EESS exhibited no neuroprotective effects in the rat model of AD. One study showed that the blank liposome exhibited no protective effects in rat model of subarachnoid hemorrhage [50]. On the other hand, our results revealed that EESS-liposome improved the neuroprotective effects by attenuating memory impairment induced by Aβ_1–42_. Consistent with our findings, liposome-based drug delivery is a potential treatment [51]. Therefore, this could indicate that the liposome could improve the delivery and neuroprotective effects of EESS in a rat model of AD by reduced learning and memory impairment.

## 5. Conclusions

The present study provided evidence that the neuroprotective effects of EESS could be improved by liposomal encapsulation. In addition, the beneficial effects of an EESS-loaded liposome expressed by improving memory ability in Aβ-induced memory impairment in rats. The proposed activity of the EESS-loaded liposome was through promoting the expression of learning and memory-related proteins BDNF/TrkB as well as synaptic proteins GAP-43 and PSD-95 in the cortex and hippocampus via p-Erk1/2/Erk1/2. These findings demonstrated that the EESS-loaded liposome could be a promising strategy for AD protection.

## Figures and Tables

**Figure 1 foods-11-02673-f001:**
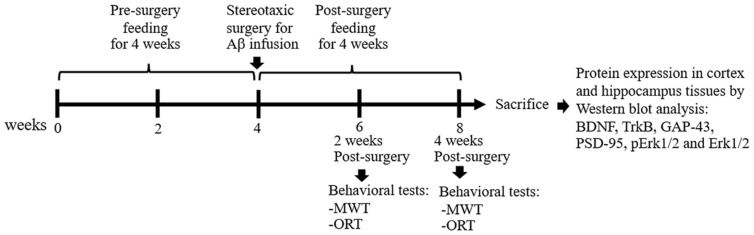
Sequence of the study procedure.

**Figure 2 foods-11-02673-f002:**
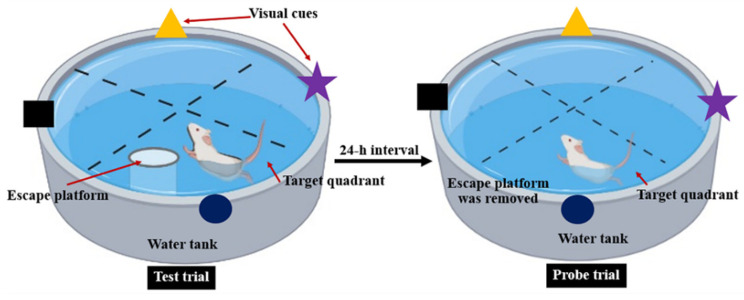
Representative procedure for Morris water maze test.

**Figure 3 foods-11-02673-f003:**
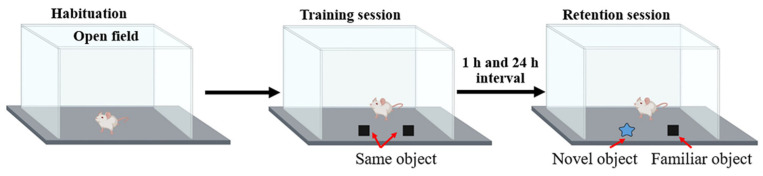
Representative procedure for object recognition test.

**Figure 4 foods-11-02673-f004:**
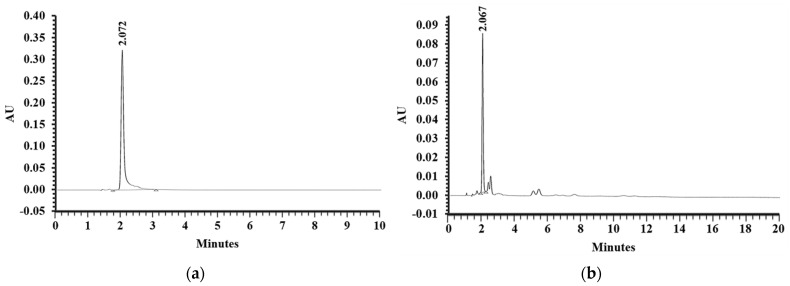
The chromatogram of (**a**) AST standard and (**b**) EESS.

**Figure 5 foods-11-02673-f005:**
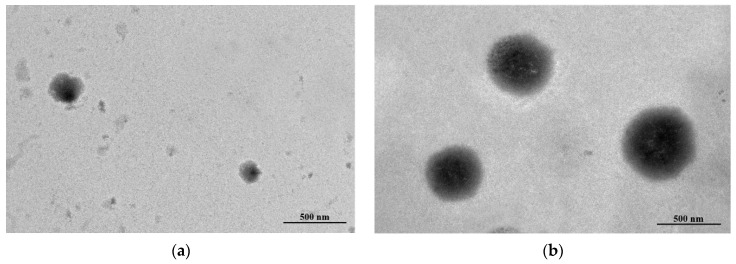
The TEM image of (**a**) blank liposome and (**b**) EESS-loaded liposome (scale bar: 500 nm and magnification: 20,000×).

**Figure 6 foods-11-02673-f006:**
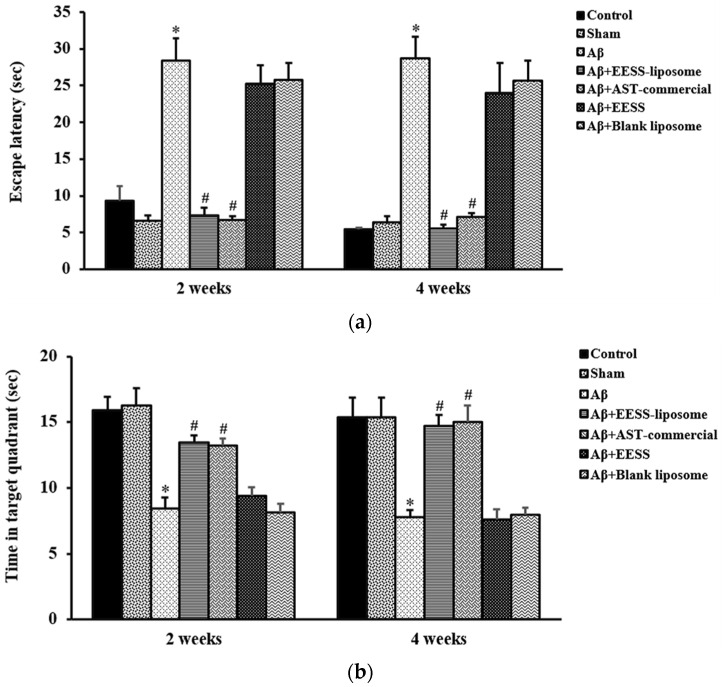
EESS-loaded liposome improved spatial memory in the MWT. (**a**) Escape latency (sec) at 2 and 4 weeks after Aβ infusion and (**b**) time in target quadrant (sec) at 2 and 4 weeks after Aβ infusion. Data were presented as means ± SEM. * indicates significant difference (*p* < 0.05) between Aβ and sham groups; # indicates significant difference (*p* < 0.05) between EESS-loaded liposome, AST-commercial and Aβ groups; *n* = 6 rats/group.

**Figure 7 foods-11-02673-f007:**
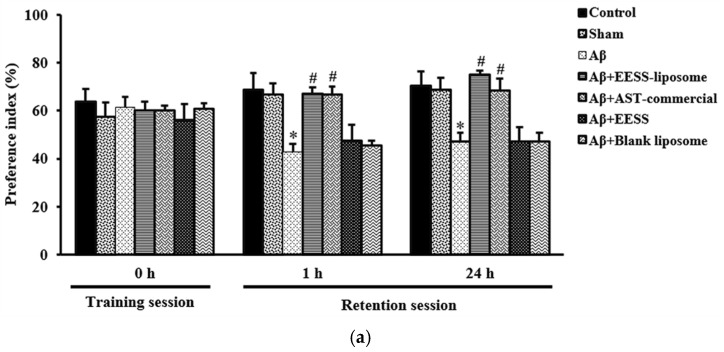
EESS-loaded liposome improved cognitive ability in the ORT. (**a**) PI of 2 weeks after Aβ infusion and (**b**) PI of 4 weeks after Aβ infusion. Data were presented as means ± SEM. * indicates significant difference (*p* < 0.05) between Aβ and sham groups; # indicates significant difference (*p* < 0.05) between EESS-loaded liposome and AST-commercial and Aβ groups; *n* = 6 rats/group.

**Figure 8 foods-11-02673-f008:**
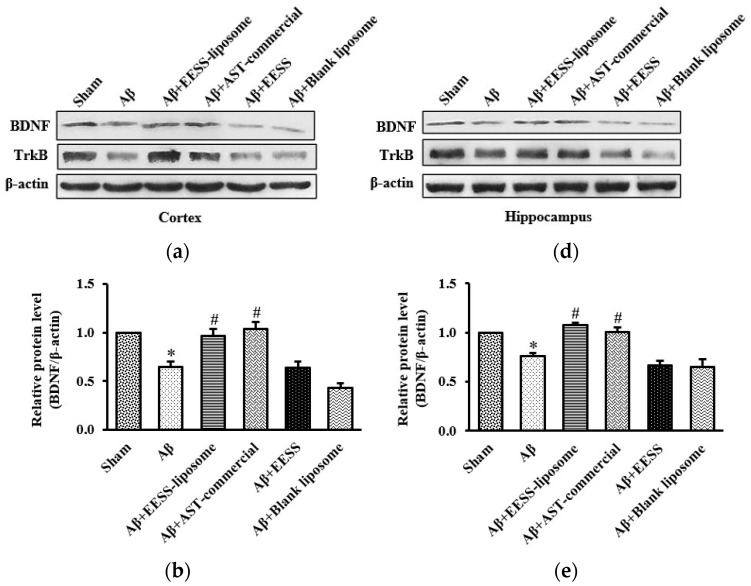
EESS-loaded liposome increased the levels of BDNF and TrkB in AD rats. (**a**,**d**) show representative images of Western blot analysis of BDNF and TrkB in the cortex and hippocampus, respectively. (**b**,**e**) show the relative BDNF levels in the cortex and hippocampus, respectively. (**c**,**f**) show the relative TrkB levels in the cortex and hippocampus, respectively. Data were presented as means ± SEM. * indicates significant difference (*p* < 0.05) between Aβ and sham groups; # indicates significant difference (*p* < 0.05) between EESS-loaded liposome, AST-commercial and Aβ groups; *n* = 5 rats/group.

**Figure 9 foods-11-02673-f009:**
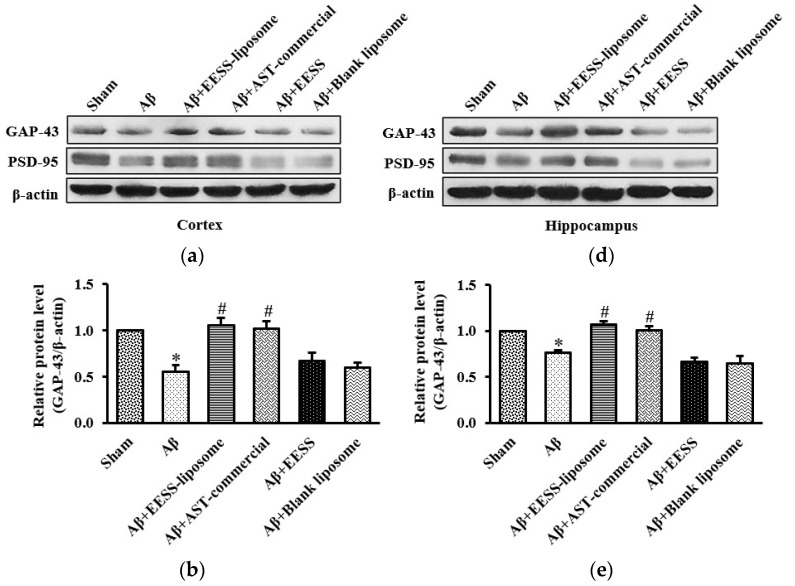
EESS-loaded liposome ameliorated Aβ-suppressed GAP-43 and PSD-95 synaptic plasticity proteins. (**a**,**d**) show representative images of Western blot analysis of GAP-43 and PSD-95 in the cortex and hippocampus, respectively. (**b**,**e**) show the relative GAP-43 levels in the cortex and hippocampus, respectively. (**c**,**f**) show the relative PSD-95 levels in the cortex and hippocampus, respectively. Data were presented as means ± SEM. * indicates significant difference (*p* < 0.05) between Aβ and sham groups; # indicates significant difference (*p* < 0.05) between EESS-loaded liposome, AST-commercial and Aβ groups; *n* = 5 rats/group.

**Figure 10 foods-11-02673-f010:**
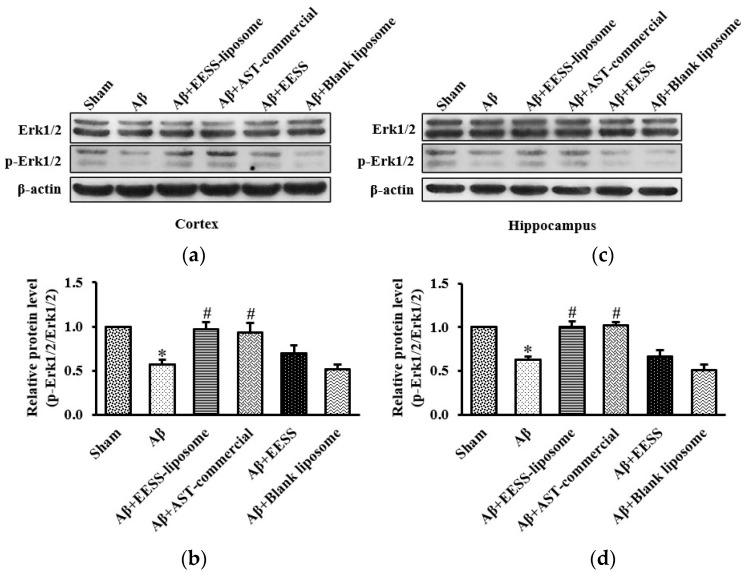
EESS-loaded liposome suppressed Aβ-induced cognition and synaptic plasticity deficits via Erk signaling pathway. (**a**,**c**) show representative images of Western blot analysis of p-Erk1/2/Erk1/2 in the cortex and hippocampus, respectively. (**b**,**d**) show the relative p-Erk1/2/Erk1/2 levels in the cortex and hippocampus, respectively. Data were presented as means ± SEM. * indicates significant difference (*p* < 0.05) between Aβ and sham groups; # indicates significant difference (*p* < 0.05) between EESS-loaded liposome, AST-commercial and Aβ groups; *n* = 5 rats/group.

**Table 1 foods-11-02673-t001:** The experimental groups and treatment of each group.

Experimental Group	Treatment
1. Control	Propylene glycol (PG; a vehicle of EESS)
2. Sham	PG + artificial cerebrospinal fluid (aCSF) ^a^
3. Amyloid-beta	PG + Aβ ^a^
4. EESS-liposome	EESS-loaded liposome 100 mg/kg BW + Aβ ^a^
5. AST-commercial	AST-commercial 10 mg/kg BW + Aβ ^a^
6. EESS	EESS 100 mg/kg BW + Aβ ^a^
7. Blank liposome	Blank liposome + Aβ ^a^

^a^ i.c.v. infusion.

**Table 2 foods-11-02673-t002:** Encapsulation efficiency, particle size, polydispersity index and zeta potential of liposome prepared using thin-film hydration method loaded with ethanolic extract from shrimp shell. Data were expressed as mean ± SD (*n* = 3).

Sample	Entrapment Efficiency (%)	Particle Size (nm)	Polydispersity Index	Zeta Potential (mV)
Blank liposome	-	587.2 ± 12.9	0.40 ± 0.00	−55.12 ± 0.61
EESS-loaded liposome	55.34 ± 0.75	1009.40 ± 38.8	0.40 ± 0.01	−61.27 ± 0.81

## Data Availability

Data is contained within the article.

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
