# Peer review of "Oral Administration of Ethanolic Extract of Shrimp Shells-Loaded Liposome Protects against Aβ-Induced Memory Impairment in Rats"

_foods, 2022, doi:10.3390/foods11172673_

Round 1

Reviewer 1 Report

The paper entitled ‘Oral administration of ethanolic extract of shrimp shells loaded liposome protects against Ab-induced memory impairment in rats’ provided an evidence that the neuroprotective property of the ESSE-loaded liposome could be a promising strategy for AD protection. On the whole, the paper is well prepared, interesting and the issue is suitable for Foods.  Please consider the following suggestions:

Abstract: informative and brief

Introduction: all necessary information was included. References are adequate and up-to-date.

Materials and methods: authors decided to use various methods (i.e. extraction of EESSand RP-HPLC analysis, preparation of EESS-loaded liposome and their characterization) in order to succeed the aim of studies. The methods are well described. In vivo studies are well designed. Methodology is adequate to subject of studies.

Statistics: provided

Results: Authors presented results in form of informative tables and figures. All obtained studies results were presented in adequate form. Results are well presented and the form is suitable for scientists from various field of pharmaceutical and medical fields. Authors presented the most important study results which are discussed in next part of paper.

Discussion: the subsection is comprehensive and include discuss based on all studies results. The discussion includes necessary references.

Conclusions: this subsection should be extended.

References: adequate and up-to-date

The subsection conclusions should be extended.    

Reviewer 2 Report

Many researchers focus on the accumulation of Aβ as the major mechanism of Alzheimer’s disease after the hypothetical thesis of amyloid (Lesne et al., 2006, Nature, 440, 6). Recently, this hypothesis has sparked controversy in the medicinal industries. Although your target protein (Aβ42) is different from Lense’s target protein (Aβ56), Is this hypothesis still available? What do you think about amyloid accumulation in relation to Alzheimer’s disease?

The ethanol extract of shrimp shells (100 mg/kg) showed no protective effect in Aβ-treated rats, but EESS-liposome at the same dose significantly improved memory ability. The difference of the memory ability is caused by improved drug delivery or the antioxidative effect of liposomes?

You indicate that the beneficial effects on learning behavior are probably through the function of BDNF/PERK1/2/Erk1/2 signaling pathway and upregulate synaptic proteins. Why?

Line 82: Generally, carotenoids were detected at 450 nm in the hplc analysis. Does astaxanthin have a specific absorption maximum at 480 nm? These carotenoids are characterized by specific absorption between 429-477 nm in vitro.

Line 207: The RSD value of AST content is over 5%. The RSD value of retention time and content must be below 5% in the validation of the analysis of chemical compounds.

Fig.5: I can’t read the words below Fig. Please revise them clearly.

Line 371: Liposome protects the degradation of AST from oxidation by heat and light. However, the liposome improves the cell penetration of substances and is therefore usually used to deliver drugs in medicine.

Line 451: What is synaptic protein? Is it five synaptic proteins, including synapsin-Ia, synapsin-Ib, snapsin-IIa, synapsin-IIb and synapsin-IIIa?
